# The Different Strategies for the Radiolabeling of [^211^At]-Astatinated Radiopharmaceuticals

**DOI:** 10.3390/pharmaceutics16060738

**Published:** 2024-05-30

**Authors:** Jie Gao, Mei Li, Jingjing Yin, Mengya Liu, Hongliang Wang, Jin Du, Jianguo Li

**Affiliations:** 1China Institute for Radiation Protection, National Atomic Energy Agency Nuclear Technology (Nonclinical Evaluation of Radiopharmaceuticals) Research and Development Center, CNNC Key Laboratory on Radiotoxicology and Radiopharmaceutical Preclinical Evaluation, Taiyuan 030006, China; jiegao1989@163.com (J.G.); 15034167070@139.com (M.L.); yjingjing1982@163.com (J.Y.); liumengya728@163.com (M.L.); 2China Institute of Atomic Energy, Beijing 102413, China; dujin@circ.com.cn; 3First Hospital of Shanxi Medical University, Taiyuan 030001, China; hongliang0812@163.com; 4China Isotope & Radiation Corporation, Beijing 100089, China

**Keywords:** astatine, radiolabeling, radiopharmaceuticals, targeted alpha therapy

## Abstract

Astatine-211 (^211^At) has emerged as a promising radionuclide for targeted alpha therapy of cancer by virtue of its favorable nuclear properties. However, the limited in vivo stability of ^211^At-labeled radiopharmaceuticals remains a major challenge. This review provides a comprehensive overview of the current strategies for ^211^At radiolabeling, including nucleophilic and electrophilic substitution reactions, as well as the recent advances in the development of novel bifunctional coupling agents and labeling approaches to enhance the stability of ^211^At-labeled compounds. The preclinical and clinical applications of ^211^At-labeled radiopharmaceuticals, including small molecules, peptides, and antibodies, are also discussed. Looking forward, the identification of new molecular targets, the optimization of ^211^At production and quality control methods, and the continued evaluation of ^211^At-labeled radiopharmaceuticals in preclinical and clinical settings will be the key to realizing the full potential of ^211^At-based targeted alpha therapy. With the growing interest and investment in this field, ^211^At-labeled radiopharmaceuticals are poised to play an increasingly important role in future cancer treatment.

## 1. Introduction

Targeted radionuclide therapy (TRT) employing alpha-emitting isotopes combined with disease-specific targeting vectors is favorable as it delivers a highly concentrated dose to a tumor site, either directly to the tumor cells or to its micro-environment, while concurrently reducing harm to the healthy surrounding tissues [1,2]. Given this potential, the development of alpha-emitting radiopharmaceuticals for targeted alpha therapy (TAT) is a vibrant field of investigation in both academia and commercial research worldwide. Several candidate isotopes for TAT are being evaluated in clinical and preclinical settings, including 223Ra, 225Ac, ^227^Th, ^213^Bi, ^212^Bi, ^212^Pb, ^211^At, ^149^Tb, and ^224^Ra. For instance, the first alpha-particle emitter, Xofigo (^223^RaCl_2_) approved by the Food and Drug Administration (FDA) in 2013 is a calcium-mimetic isotope that is incorporated into sites of increased bone turnover and osteoblastic activity [3,4]. Targeted ^225^Ac and ^227^Th conjugates, unlike ^223^Ra, signify a novel class of therapeutic radiopharmaceuticals for TAT [5,6,7]. For example, ^225^Ac-DOTATATE is a promising treatment option that introduces a new dimension in patients who are unresponsive to ^177^Lu-DOTATATE therapy or have attained the maximum prescribed cycles of ^177^Lu-DOTATATE therapy [8]. However, the supply of α-particle-emitting radionuclides (such as ^225Ac^) and the radiotoxic daughter products from decay (such as ^213^Bi) is considered as a potential obstacle and challenge for the growth of the field [9].

^211^At is a promising candidate isotope for TAT due to its chemical properties suitable for labeling targeting vectors, and the emission of just a single α-particle per decay, which offers enhanced control over off-target effects, although its 7.2 h half-life is relatively short [1]. One branch occurs via electron capture and leads to ^211^Po, which subsequently decays to stable ^207^Pb by emitting an α-particle. This process generates a 77–92keV polonium x-ray, providing targeting strategies using this α-particle emitter with a theranostic characteristic for non-invasive nuclear planar or SPECT of real-time biodistribution [10,11]. The other decay branch leads to ^207^Bi through alpha-particle emission, with a half-life of 33.9 years, and decays to ^207^Pb by electron capture. These decay pathways result in 100% alpha-particle emission during the decay of ^211^At (5.87 and 7.45 MeV in 41.8% and 58.2% of the decays, respectively) [12] (Figure 1).

More significantly, in terms of the preparation and supply of ^211^At, significant challenges also exist, which are distinct from those encountered with the other alpha nuclides (^225^Ac and ^227^Th). The primary method for ^211^At production involves irradiating natural bismuth with α-particles via the ^209^Bi(α,2n)^211^At nuclear reaction by cyclotron [13], and it can also be produced via the generator with ^209^Bi(^7^Li,5n)^211^Rn reaction, where ^211^Rn decays to ^211^At, but this method is currently under development [14]. Encouragingly, the requirement for improved accelerator infrastructure to produce ^211^At recently has prompted concrete action from government agencies across various regions, including the US, EU, and Japan [15]. For example, five sites have implemented the routine production and purification of ^211^At, resulting in a total of six production sites of ^211^At, enabling many preclinical and clinical uses of ^211^At in Japan [16,17]. Sadahiro Naka et al. [18] successfully established a stable method of [^211^At]NaAt solution under GMP compliance that can be administered to humans intravenously as an investigational product. Europe and the US address the needs of ^211^At through a funded multi-institutional program and enhanced cyclotron performance, respectively.

It is worth noting that ^211^At has the capability to radiolabel small molecules and monoclonal antibodies (mAbs), encompassing thymidine analogs, biotin analogs, and bisphosphonate complexes, a potential advantage compared with radiometals, which require somewhat bulky polydentate ligands for stable incorporation. Until now, multiple ^211^At-astatinated clinical trials have been conducted [19]. The initial study was released by the Zalutsky Group at Duke University in 2008 [20]. ^211^At-ch81C6 was used to treat residual disease following the surgical resection of the glioma in the brain, with a total of 96.7% ± 3.6% of ^211^At decayed in the surgically created resection cavity (SCRC), and the mean blood pool percentage injection dose was ≤0.3%ID/g. The outcomes were highly encouraging overall, showing a median overall survival time increase from 31 weeks to 54 weeks with additional ^211^At-ch81C6, and no observed dose-limiting toxicity. In 2009, the second clinical trial (phase I) was reported, focusing on treating recurrent ovarian cancer after second-line chemotherapy with ^211^At-MX35 F(ab’)_2_ [21]. The decay-corrected activity concentration in the thyroid rose to 127 ± 63% of the initial activity concentration (IC) at 20h without blocking and remained below 20% IC with blocking. There were no detectable uptakes in other organs, and neither subjective nor laboratory adverse effects were observed. Until 2019, Andreas Hallqvist et al. [22] provided a long-term follow-up on intraperitoneal α-emitting radioimmunotherapy using ^211^At in relapsed ovarian cancer. Among the four patients who survived for more than 6 years, one did not experience relapse, and no evident signs of radiation-induced toxicity or decreased tolerance to relapse therapy were observed. In addition, Yawen Li et al. [23] reported an astatine-211(^211^At)-labeled anti-CD45 monoclonal antibody, ^211^At-BC8-B10, of interest for clinical trials involving allogeneic hematopoietic transplantation in the treatment of advanced hematological malignancies. This study led to the initiation of a phase I/II clinical trial using ^211^At-BC8-B10 by the FDA. Shyril O Steen et al. conjugated ^211^At to an anti-CD38 monoclonal antibody to develop an ^211^At-CD38 therapy in a planned clinical trial [24]. Hiroaki Echigo et al. developed probes for multiradionuclide radiotheranostics using RGD peptide Ga-DOTA-[^211^At]c [RGDf(4-At)K] for clinical applications. [25,26]

Unfortunately, the vast majority of the ^211^At-labeled compounds of interest have been halted owing to the lack of in vivo stability of these compounds, although there are different reasons for instability than the other alpha-radiometals [27,28,29]. The premature release of unbound ^211^At in vivo has been observed, which can result in the irradiation of nontarget tissues, including physiological accumulation in the thyroid and stomach; the loss of dissociation and off-target of the radiometal is associated with therapeutic toxicity. To avoid toxic effects, highly stable ^211^At-labeled compounds must be developed for further study. In this review, with ^211^At, the different strategies for the radiolabeling procedures were described, aiming to play a catalytic role in the promotion of the research and development of [^211^At]-astatinated radiopharmaceuticals.

## 2. General Strategies for ^211^At Radiolabeling

In general, the satisfactory labeling method of ^211^At requires adequate yields, modest conditions (temperature, solvent, oxidizing agent, and so on), short reaction time, and so on. The procedure must also maintain the specificity and affinity of the labeled carrier to its molecular target, and the bond between the ^211^At and the molecule must remain stable in vivo. Iodine and ^211^At are from the same group with similar chemical properties; in fact, ^211^At also exhibits significant metallic characteristics. An important consideration is the carbon–halogen bond strength, which is lower for astatine at 49 kcal/mole compared to iodine at 62 kcal/mole [30]. What is worse, the absence of the stable isotopes of astatine complicates the synthetic and analytical chemistry of ^211^At-labeled compounds, which has significant restrictions on the research of ^211^At radiopharmaceuticals [11].

Basically, the general synthetic approaches of radio-astatination corresponding to various carriers can be investigated through either nucleophilic or electrophilic reactions, involving the formation of aryl astatine–carbon bonds. The classification of strategies for radiolabeling ^211^At radiopharmaceuticals: nucleophilic substitution occurs with halogen anions including halogen exchange, dediazoniation, and boron astatine, and electrophilic substitution occurs with electropositive halogens including direct aromatic substitution and demetallation reactions. The applications of these approaches will be discussed in the sections below (Figure 2).

## 3. Nucleophilic Substitution

### 3.1. Halogen Exchange

This reaction is typically conducted on iodinated derivatives, with examples provided for substitution on alkyl carbons and aromatic nucleophilic. Lui B et al. [31] have found that the halogen exchange labeling of 6β-iodomethyl-19-norcholest-5(10)-en-3β-ol (NCL-6-I) with ^211^At using a crown ether as a medium enables significantly shorter reaction times compared to previously reported exchange methods using organic solvents. This process, initiated from the iodinated derivative, can be completed in 10 min at 70 °C, yielding about 80%. But, the radioactivity is observed in the thyroid tissue with NCL-6-^211^At ranging from 3 to 5% of the administered dose at 6–12 h, indicating the in vivo deastatination of the administered steroid because of the weakness of the Calkyl-At bond. Phenylalanine, targeting L-type amino acids transporter 1 (LAT1), is highly expressed in various types of human cancer, may prove useful for a broad range of cancers, and has been previously labeled with ^211^At using different methods. Meyer et al. [32] synthesized 2-[^211^At]-L-phenylalanine and 4-[^211^At]-L-phenylalanine from the corresponding iodo and bromo derivatives, marking the first application of Cu+-catalyzed nucleophilic halogen exchange reactions for astatination. The radiochemical yields of 2-[^211^At]-L-phenylalanine and 4-[^211^At]-L-phenylalanine by nucleophilic exchange ranged from 52–74% to 65–85% with HPLC purification. Notably, this reaction pathway necessitates high temperatures for completion, and the separation of the radiohalogenated product from a chemically similar substrate has often posed a challenge. Although the nucleophilic halogen exchange methods increase the rapidity of the reaction and provide adequate yields for experiments, their use is limited to substrates capable of withstanding relatively harsh conditions due to the high temperatures required.

### 3.2. Dediazoniation

Diazonium salts find extensive application in organic synthesis for the derivatization of aromatic rings, including halogenations. In 1977, Friedman A.M et al. [33] first reported a two-step process forming non-labile At-protein bonds. A stable astatine compound, p-astatobenzoic acid, was first prepared via a diazonium salt intermediate, and this compound was then added to bovine serum albumin (BSA) by a condensation reaction between the carboxylic acid group and an amine function on the protein. The results suggest that the astatinated protein is stabilized in vivo over a period of 20h. The studies of immunodiffusion and haemagglutination indicated that the ^211^At-labelled BSA possessed the majority of the antigenic properties of unlabeled BSA. Visser G W M et al. [34] demonstrated that astatine reacts preferentially with the type of aromatic diazonium salt that decomposes via a radical reaction channel, and obtained compounds with ^211^At (up to 90%) by heating at 50 °C until the end of nitrogen evolution. Wunderlich G et al. [35] developed a simple and rapid method for labeling proteins with ^211^At using a 1,4-diaminobenzene link. 1,4-diaminobenzene is transformed into the diazonium salt, allowing for the simultaneous reactions of both ^211^At and proteins with the diazonium salt. In the presence of a protein, ^211^At reacted with the bisdiazonium at 20 °C for 1 h, resulting in yields of about 50–55%. Due to the high dilution of astatine involved, achieving a satisfactory radiochemical yield via the approach appears unlikely. In fact, while reactions involving radioiodine have been demonstrated to function, they typically result in low yields (max 15%). According to Meyer et al. [36], the latter seems more likely, between heterolytic cleavage resulting in the formation of an aryl cation and homolytic cleavage resulting in an aryl radical. Due to its higher polarizability, astatine exhibits a greater tendency to form a stable complex with the diazonium than iodine. Similarly, astatine also presents a greater inclination to cede an electron to form a radical capable of recombining with the aryl radical in the following sequence [37]. Notably, the preparation of the diazonium necessitates harsh conditions (oxidative and acidic media), thus restricting the method to less-sensitive substrates. Meanwhile, a high proportion of side-products, often precipitated in the medium, complicates the purification and isolation of the product. To avoid the competing reaction of water with the diazonium resulting in the phenol derivative, an excess of halogen is necessary.

### 3.3. Boron-Astatinated

Recent reports have highlighted promising nucleophilic strategies utilizing the radiohalogen as a nucleophilic species (X^−^) based on arylboronic acids/ester precursors that provided high radiochemical yields on a wide range of substrates [38]. Related research indicated that the use of boron clusters such as decaborate, dodecaborate, and carborane has been considered recently to enhance the stability of antibodies labeled with ^211^At [39,40], which are more stable than those linked by aromatic ones [41,42].

Sean W. Reilly et al. [43,44] have demonstrated the first method to astatinate compounds with boronic ester precursors, providing an efficient and non-toxic protocol that eliminates the conventional reliance on toxic organotin reagents. They prepared an [^211^At] PARP inhibitor in high radiochemical yields by employing the corresponding boronic ester precursor and [^211^At]NaAt in the presence of Cu(pyridine)4OTf in a 4:1 mixture of methanol and acetonitrile. Shirakami et al. [45] devised a novel approach for astatination by the substitution of ^211^At for a dihydroxyboryl group coupled to phenylalanine. They obtained [^211^At]4-astato-L-phenylalanine ([^211^At]APA) as the carrier-free product in aqueous medium in high radiochemical yields (98.1 ± 1.9%, n = 5). The crude reaction mixture underwent purification via solid-phase extraction, resulting in a radiochemical purity of 99.3 ± 0.7% (n = 5). The high yield and purity were attributed to the formation of [^211^At]AtI and AtI_2_^-^ as the reactive intermediates in the astatination reaction. This reaction did not necessitate the use of organic solvents or toxic reagents, indicating its suitability for clinical applications.

This method could be successfully applied for the astatination not only of small molecules but also of biomolecules, such as antibodies. To overcome issues to eliminate the need for toxic organotin agents, Marion Berdal et al. [46] explored the potential of arylboron chemistry as an alternative method for the late-stage labeling of aBA-9E7.4, a CD138-targeting monoclonal antibody (mAb). The reactivity of a model precursor, 4-chlorobenzeneboronic acid with nucleophilic ^211^At, was at first investigated in aqueous conditions. In the presence of a copper (II) catalyst and 1,10-phenanthroline, radiochemical yields >95% were attained within 30 min at room temperature. DS Wilbur et al. [47] intended to determine whether the nido-carborane-containing biotin derivatives provide a significant improvement in astatine stability over the biotin derivatives previously studied, although the biodistribution studies indicated that deastatination was still occurring. Katsumasa Fujiki et al. [48] report a facile synthesis of an ^211^At-labeled trastuzumab (anti-HER2 antibody) by designing and synthesizing a tetrazine probe having closo-decaborate(2−) as the prosthetic group for binding ^211^At, which forms a bioavailable stable complex. The one-pot three-component double-click labeling method could be utilized without reducing the antibody binding affinity, but the ^211^At decaborate complex proved unsuitable as an intravenously administered antibody-based therapeutic due to the significant hydrophobic properties of the labeling agent which may have altered the native biodistribution of the antibody.

While the boron cage derivatives provided stable ^211^At-labeled compounds against in vivo deastatination, several experiments indicated that the borane derivatives may influence the absorption, distribution, and elimination of the carriers, making corresponding radiopharmaceuticals show longtime retention in the blood, kidney, or liver [49]. Meanwhile, it is also not applicable to radioiodination due to its reliance on electrophilic halogen species, which may once again react with tyrosines and histidine residues. These results call for a new scaffold that provides ^211^At-labeled compounds of high stability against in vivo deastatination. Hiroyuki Suzuki et al. [50] confirmed that ^211^At-labeled neopentyl glycol also remained stable against both nucleophilic substitution and CYP-mediated metabolism. The biodistribution profiles of ^211^At-labeled neopentyl glycol resembled those of its radioiodinated counterpart in contrast to the ^125^I/^211^At-labeled benzoate pair. Urine analyses confirmed the excretion of ^211^At-labeled neopentyl glycol in the urine as a glucuronide conjugate with no free At^−^. These findings suggest that neopentyl glycol would constitute a promising scaffold to prepare a radiotheranostic system with ^211^At.

## 4. Electrophilic Substitution

### 4.1. Direct Aromatic Electrophilic Substitution

Direct astatination using electrophilic ^211^At (utilizing chloramine-T or H_2_O_2_ as oxidizing agent), although it had no denaturation of the protein, was demonstrated to be unstable in vivo due to the release of free ^211^At [51]. Initially, the instability was ascribed to the vulnerability of the At–C [52] or At–S bond (this kind of bonding is easily hydrolyzed in vivo) [53], assuming that astatine reacted with the tyrosine, histidine, or cysteine residues of the proteins when employing the direct radiolabeling method. Norseyev Y et al. [54] completed the carrier-free astatotyrosine using an electrophilic reaction in acidic media, and the yield was about 90%. The optimal conditions of a temperature range of 150–160 °C and reaction time of 20–30 min were chosen for synthesizing astatotyrosine that was denaturing for proteins, which means it was performed with radioiodine and was unlikely with astatine. Consequently, astatine could not be used in the direct radiolabeling procedures of proteins, necessitating the development of prosthetic groups for indirect radiolabeling analogous to those developed for radioiodination [37]. But there is always an exception, so a method to produce ^211^At-methylene blue has been devised [55]. The conditions for transferring astatine to a chemically active electrophilic form were found. The method provides the radiochemical yield of ^211^At-methylene blue as large as 68 ± 6% in 15 min at 100 °C, with sodium persulfate as the oxidizing agent. However, due to the substantial starting material required, the specific activity remains very low, and purification is necessary.

In 2016, Brechbiel et al. [56] initially assumed that the nucleophilic substitution of astatide on aryliodonium salts would proceed similarly to iodide. However, this study revealed how regioselectivity was driven by electronic effects, as ^211^At-product formation proceeded with high selectivity for the aryl ring containing the most electron-withdrawing substituent in the para-position. These findings highlight the utility of diaryliodonium salts much more than expected in synthesizing astatinated compounds, particularly with the radioisotope ^211^At. Despite achieving high yields with this approach, the astantination of the conjugate aryl group leads to the formation of unwanted side-products.

### 4.2. Demetallation Reactions

Electrophilic demetallation, the most versatile astatination strategy, is the utilization of an astatodemetallation reaction with a tin or silicon precursor, from small molecules to biomolecules. Organomercuric compounds were the initial precursors in search which enabled the introduction of ^211^At with high yields into various compounds. The radio-astatination of a diverse range of organic compounds with organomercury precursors has been documented by Visser and co-workers [57,58]. Nonetheless, the inclusion of the highly toxic traces of mercury in the resultant compound has hindered their application for pharmaceutical use. Metals from group IV of the periodic table, particularly tin, silicon, and boron, have been preferred. For example, in initial studies, Milius R A et al. [59] employing trialkylstannyl precursors necessitating the use of iodide carriers to achieve reasonable ^211^At labeling yields. More recently, astatination via electrophilic destannylation without the use of an iodide carrier has been reported [60,61]. Like [^131^I]MIBG, Vaidyanathan G et al. [62] have developed two steps for the preparation of [^211^At]MABG from 3-(tri-n-butylstanny1) benzylamine. In the limited experiments conducted, the overall yield was 13% because of the considerable water content in the activity, and this method involved an extra step after the introduction of ^211^At, whereas the research team developed a one-step approach for the preparation of [^211^At]MABG from 1-[3-(trimethylsilyl)-benzyllguanidine, and the yields were 87.7 ± 4.3% at 70 °C in the presence of trifluoroacetic acid as the medium and N-chlorosuccinimide as the oxidant. Considering that organotin compounds must be separated efficiently from the desired astatinated compound, like the method using a precursor grafted on a polymer support that simplifies the process of purification and significantly reduces the quantity of liberated tin, Vaidyanathan G et al. [63] developed a convenient approach to synthesize [^211^At]MABG at a high level that is capable of kit formulation. A tin precursor anchored to a solid support was treated with a methanolic solution of ^211^At with a mixture of acetic acid as the oxidant, which led to radiochemical yields of 63 ± 9% within a shorter synthesis duration. The radiochemical purity exceeded 90% with no detected chemical impurity. Ana P. Kiess et al. [64] developed a urea-based, ^211^At-labeled small molecule(^211^At-6)-targeting prostate-specific membrane antigen (PSMA) for the treatment of micrometastases induced by prostate cancer via N-succinimidyl-4-tributylstannyl benzoate, in the presence of acetic acid and N-chlorosuccinimide, at the room temperature 30min, astatodestannylation, and the in situ deprotection of precursor delivered ^211^At-6 in 62.6 ± 9.5%. Due to theoretical advantages, many preclinical studies amply demonstrated that [^211^At]PSMA or close analogs hold promise for translation in the treatment protocol of mCRPCa patients, which for sure is a giant leap towards cost-effective precision oncology [65].

Although tin precursors are readily introduced on a wide range of compounds through well-established synthetic methods, the handling of toxic alkyltin reagents is necessary for developing aryl stannane precursors. Furthermore, the low acid-resistance of stannyl precursor requires the use of an N- and C-terminus-protected precursor, leading to the modest yields of the ^211^At-radiolabeled products due to the involvement of multiple synthetic steps. This is attributed to the astatine’s ability to adopt multiple oxidation states, therefore, making it difficult to obtain the desired, and non-stable, At+1 species for electrophilic substitution. Watanabe et al. [66] reported the synthesis of 4-[^211^At]-L-phenylalanine ([^211^At]APA) via electrophilic desilylation, where the corresponding ^211^At-labeled 4-triethylsilyl-L-phenylalanine-substituted precursor was heated at 70 °C for 10 min, showing competitive radiochemical yield (65–85%) and radiochemical purity (>99%) (Table 1).

The recent clinical advancements in targeting therapies via attaching them to antibodies that target tumor-restricted biomarkers mark a new era in cancer treatments. However, despite their promise, these therapies may face limitations in completely eradicating tumors due to their lots of shortcomings. One highly promising strategy to overcome this challenge is to target potent radioactive isotopes specifically to tumors by molecular delivery systems like antibodies and derivatives [70]. ^211^At offers many attractive features for targeted radiotherapy, such as compatibility with various carriers capable of delivering a dose to patients.

In recent years, the primary approach for radiolabelling proteins with ^211^At involves a two-step procedure based on the radiosynthesis of an intermediate prosthetic group N-[^211^At]succinimidylastatobenzoate ([^211^At]SAB), by the electrophilic astatodestannylation of the corresponding aryltrialkylstannane precursor, similar to the analogous procedure that had been developed previously for radioiodination [71]. Emma Aneheim et al. [72] indicated that immunoconjugates formed by conjugating an antibody (e.g., Traztuzumab) with alkyltin-based prosthetic groups (N-succinimidyl 3-(trimethylstannyl)benzoate) can be stored in a neutral buffer (pH 7.4) for over 3 months at 4 °C and they can be stored at −20 °C without compromising the quality of the labeled product. Lindegren et al. [73,74] developed a single-step strategy from the precursor grafted to the antibody prior, thereby saving time and enhancing yields compared to two-step approaches. Nevertheless, it still relies on the established electrophilic destannylation reaction. Labeling complex molecules like proteins with heavy radiohalogens is frequently poorly effective due to the requirement for multiple steps and intermediate purifications.

As mentioned above, the ability of mAbs to target tumor-associated antigens has high affinity and specificity. However, the prolonged circulation duration of full-length immunoglobulins, i.e., they can take several days to achieve optimal biodistribution in the body, necessitates their labeling with radionuclides with multiday physical half-lives, for example, ^225^Ac (t_1/2_ 10d) are commonly used for radioimmunotherapy [75]. The unavoidable combination of long biological and physical half-lives can create high radiation dose rates to healthy tissues, a complication that has dampened enthusiasm for radioimmunoconjugates in the clinic. On the one hand, single-domain antibody fragments (sdAbs), alternatively termed VHH molecules and nanobodies, present a promising platform for selectively delivering radionuclides to cancer cells for nuclear imaging and targeted radionuclide therapy. The use of sdAbs for TRT with the ^211^At is particularly attractive due to the high alignment of the physical half-life (7.2 h) with the pharmacokinetic profile of sdAbs. Yutian Feng et al. [76] have evaluated iso-[^211^At]-AGMB-PODS via the electrophilic astatodestannylation of the corresponding precursor, which was significantly more stable in vitro compared to its maleimide analog in the presence of cysteine and human serum albumin (HSA), demonstrating excellent tumor uptake and high in vivo stability. On the other hand, in vivo pretargeting is predicated on separate injections of the immunoglobulin and radionuclide and relies upon a biorthogonal ligation to join two components together within the body. Labeling the antibody with a fast-moving, small molecule radioligand after it has achieved an optimal biodistribution in vivo, thereby limiting the circulation duration of the assembled radioimmunoconjugate in the blood. More significantly, it is the fact that this method enables the utilization of radionuclides with shorter half-lives that are normally incompatible with full-length IgG, such as ^211^At (t_1/2_ 7.2h). Chiara Timperanza et al. [77] have synthesized and evaluated poly-L-lysine-based effector molecules for pretargeting applications, leveraging the tetrazine and trans-cyclooctene reaction with ^211^At for targeted alpha therapy, and high specific astatine activity was achieved without affecting the stability of the radiopharmaceutical or the binding between tetrazine and transcyclooctene.

## 5. Conclusions and Future Perspectives

In conclusion, ^211^At-labeled radiopharmaceuticals have shown great promise for the targeted alpha therapy of cancer, owing to the favorable nuclear properties of ^211^At and the high cytotoxicity of alpha-particles. However, the widespread clinical application of ^211^At is currently hindered by the limited in vivo stability of ^211^At-labeled compounds, which can lead to off-target toxicity and reduced therapeutic efficacy. To address this challenge, significant efforts have been devoted to the development of novel bifunctional coupling agents and labeling strategies that can enhance the stability of ^211^At-labeled radiopharmaceuticals. While progress has been made in this regard, there is still much room for improvement and innovation.

Looking forward, several key areas of research are expected to drive the future development of ^211^At-based targeted alpha therapy. Firstly, the design and synthesis of new bifunctional coupling agents with improved resistance to deastatination will be crucial for enhancing the in vivo stability and targeting specificity of ^211^At-labeled radiopharmaceuticals. This may involve the use of novel chemical scaffolds, such as nanoparticles, liposomes, or engineered protein constructs, that can protect the ^211^At-labeled moiety from degradation while enabling efficient delivery to the tumor site. Additionally, the exploration of alternative labeling approaches, such as click chemistry, enzyme-mediated reactions, and supramolecular assembly, may lead to the development of more efficient and specific ^211^At-labeling methods.

Secondly, the identification of new molecular targets and carrier molecules for ^211^At-based targeted alpha therapy will be essential for expanding the therapeutic scope and clinical applicability of this approach. While antibodies and peptides have been the mainstay of ^211^At radiopharmaceuticals to date, there is growing interest in the use of smaller targeting moieties such as aptamers, affibodies, nanobodies, and engineered protein scaffolds, which offer the advantages of improved tissue penetration, faster clearance, and lower immunogenicity. Moreover, the use of multivalent and multifunctional carriers, such as nanoparticles and exosomes, may enable the delivery of higher payloads of ^211^At to the tumor site while minimizing off-target toxicity. The combination of ^211^At with other therapeutic modalities, such as chemotherapy, immunotherapy, and targeted drug delivery, may also lead to synergistic effects and improved treatment outcomes.

Thirdly, the optimization and standardization of ^211^At production and quality control methods will be critical for ensuring the reliable and consistent supply of high-quality ^211^At for clinical use. This will require the development of more efficient and automated methods for ^211^At isolation, purification, and radiolabeling, as well as the establishment of robust quality control assays for assessing the purity, specific activity, and stability of ^211^At-labeled compounds. The use of advanced analytical techniques, such as mass spectrometry and nuclear magnetic resonance spectroscopy, may provide valuable insights into the chemical and structural properties of ^211^At-labeled radiopharmaceuticals, enabling the rational design of more stable and effective compounds.

Finally, the continued preclinical and clinical evaluation of ^211^At-labeled radiopharmaceuticals will be essential for realizing the full potential of this approach. This will involve the development of standardized protocols for ^211^At production, quality control, and administration, as well as the establishment of appropriate animal models and clinical trial designs. The use of advanced imaging techniques, such as positron emission tomography (PET) and single-photon emission computed tomography (SPECT), may enable the non-invasive monitoring of ^211^At biodistribution and tumor targeting, facilitating the optimization of dosing regimens and treatment planning. Moreover, the long-term follow-up of patients treated with ^211^At-labeled radiopharmaceuticals will be crucial for assessing the potential risks and benefits of this approach, including the incidence of secondary malignancies and other late toxicities.

In summary, while significant progress has been made in the development of ^211^At-labeled radiopharmaceuticals for targeted alpha therapy of cancer, there are still many challenges and opportunities for future research. The continued innovation in ^211^At labeling chemistry, carrier molecule design, and production methods, coupled with the rigorous preclinical and clinical evaluation of new ^211^At-labeled compounds, will be the key to unlocking the full therapeutic potential of this promising approach. With the growing interest and investment in targeted alpha therapy, it is expected that ^211^At-based radiopharmaceuticals will play an increasingly important role in future cancer treatment, offering new hope for patients with advanced or metastatic disease.

## Figures and Tables

**Figure 1 pharmaceutics-16-00738-f001:**
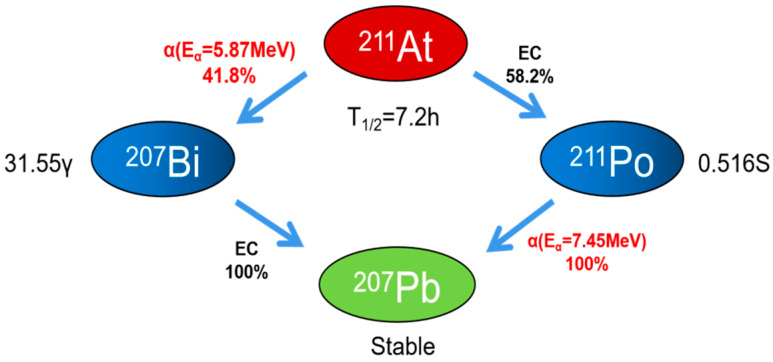
Decay scheme of ^211^At.

**Figure 2 pharmaceutics-16-00738-f002:**
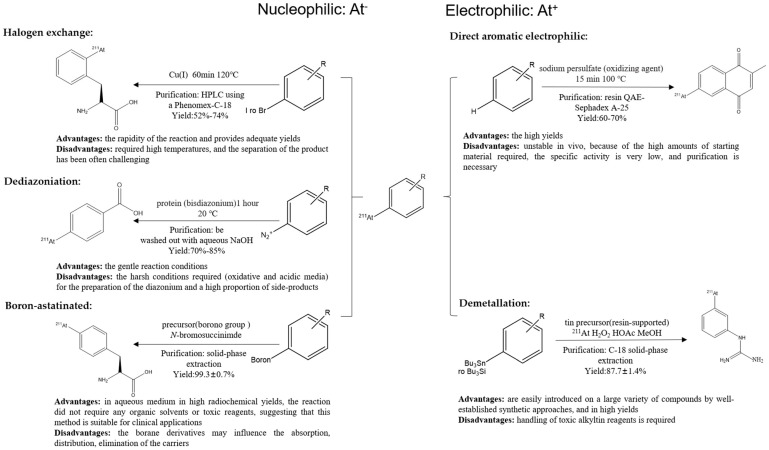
Main reaction pathways for ^211^At radiolabeling.

**Table 1 pharmaceutics-16-00738-t001:** Examples of ^211^At radiolabeling reaction strategies of small molecules.

Compounds	Structure	Strategies of Reaction	Potential in Clinical	Current Status
[^211^At]-benzylguanidine(^211^At-MABG)	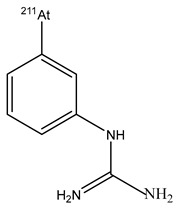	The demetallation reactions of electrophilic substitution (destannylation)	NET-associated tumors	Clinical
2-[^211^At]-L-phenylalanine	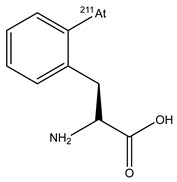	The halogen exchange of nucleophilic substitution	glioma tumor	Preclinical
4-[^211^At]-L-phenylalanine([^211^At]APA)	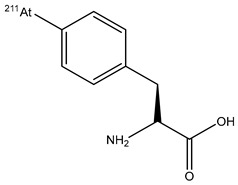	Substitution for dihydroxyboryl group (boron-astatinated)	glioma tumor and ovarian cancer	Preclinical
		The demetallation reactions of electrophilic substitution (desilation)		
α-methyl-l-tyrosine (^211^At-AAMT)	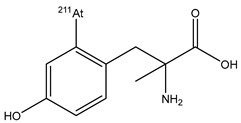	Direct aromatic electrophilic substitution	pancreatic cancer	Preclinical
4-^211^At-astato-N-[4-(6-(isopropylamino)pyridine-4-yl)-1,3-thiazol-2-yl]-N-methylbenzamide (^211^At-AITM) [67]	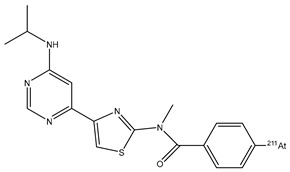	The demetallation reactions of electrophilic substitution (destannylation)	oncoprotein mGluR1 for melanoma	Preclinical
[^211^At]-(2S)-2-(3-(1-carboxy-5-(4–211At-astatobenzamido)pentyl)ureido)pentanedioic acid [68]	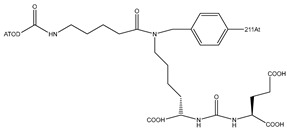	The demetallation reactions of electrophilic substitution (destannylation)	prostate cancer	Preclinical
^211^At-labelled FAPI-04 [69]	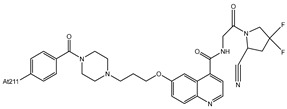	The demetallation reactions of electrophilic substitution (destannylation)	glioma tumor	Preclinical

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
