# Peer review of "The Different Strategies for the Radiolabeling of [211At]-Astatinated Radiopharmaceuticals"

_pharmaceutics, 2024, doi:10.3390/pharmaceutics16060738_

Round 1

Reviewer 1 Report

Comments and Suggestions for Authors

Comments:

1)     The review is about Astatine-211 (²¹¹At) as a promising radionuclide for targeted alpha therapy of cancer. The authors correctly recognize that the limited in vivo stability of ²¹¹At-labeled radiopharmaceuticals remains a major challenge. 

The authors provide next a rather comprehensive overview of the current chemical strategies for ²¹¹At radiolabeling, including nucleophilic and electrophilic substitution reactions, as well as the recent advances in the development of novel bifunctional coupling agents and labeling approaches to enhance the stability of ²¹¹At-labeled compounds. 

They discuss also the preclinical and clinical applications of ²¹¹At-labeled radiopharmaceuticals, including small molecules, peptides, and antibodies. 

The outlook concerns the identification of new molecular targets, the optimization of ²¹¹At production and quality control methods, and the continued evaluation of ²¹¹At-labeled radiopharmaceuticals in preclinical and clinical settings. 

2)     The review is unbiased (including avoiding too many self-citations). 

However, recent work may be improved, for instance you should discuss

3)     Makvandi M, Samanta M, ….Pryma DA, Maris JM. Pre-clinical investigation of astatine-211-parthanatine for high-risk neuroblastoma. Commun Biol. 2022 Nov 17;5(1):1260.

and not just the 2018 paper with Makvandi.

4)     Four papers from 2022, five from 2021, NONE from 2024 and with 2023, for instance look at these ones (cursory selection, is your work not my to be update and pick the right works):

1: Naka S, Ooe K, Shirakami Y, Kurimoto K, Sakai T, Takahashi K, Toyoshima A,

Wang Y, Haba H, Kato H, Tomiyama N, Watabe T. Production of

[<sup>211</sup>At]NaAt solution under GMP compliance for investigator-initiated

clinical trial. EJNMMI Radiopharm Chem. 2024 Apr 15;9(1):29. doi:

10.1186/s41181-024-00257-z. PMID: 38619655; PMCID: PMC11018728.

2: Echigo H, Munekane M, Fuchigami T, Washiyama K, Mishiro K, Wakabayashi H,

Takahashi K, Kinuya S, Ogawa K. Optimizing the pharmacokinetics of an

<sup>211</sup>At-labeled RGD peptide with an albumin-binding moiety via the

administration of an albumin-binding inhibitor. Eur J Nucl Med Mol Imaging. 2024

Apr 4. doi: 10.1007/s00259-024-06695-w. Epub ahead of print. PMID: 38570359.

Comments on the Quality of English Language

fine, only minor checks required

Reviewer 2 Report

Comments and Suggestions for Authors

This manuscript reviews the advances of diverse methods for labeling of 211At radionuclide in terms of chemistry. This is a good topic and can contribute to the related research field.

There are some suggestions that the authors can further improve the paper before publication.

Specific comments:

1.    The general chemistry was given in Fig 2. For each of the reaction type in Section 3.1, 3.2 – 4.2, the authors can provide an example of the reaction chemistry published previously.

2.    A Table comparing the advantages and disadvantages of each method is needed.

3.    Table 1, it is better to present the structure of each compound as well.

4.    The efficiency for labeling is very important, and the authors need to discuss the efficiency for each method. Also, how to remove the unreacted 211At radionuclide after labeling? This can be added where it is suitable or in the discussion part.

5.    The most recent papers are really needed to be cited and discussed as well, especially from 2022 to 2024.

Round 2

Reviewer 1 Report

Comments and Suggestions for Authors

thank you for the revisions, all is fine from my perspective